# Efficient Multimodal Dataset Distillation via Generative Models

**Zhenghao Zhao**[1]    **Haoxuan Wang**[1]    **Junyi Wu**[1]
**Yuzhang Shang**[2]    **Gaowen Liu**[3]    **Yan Yan**[1]*
[1]University of Illinois Chicago    [2]University of Central Florida    [3]Cisco Research
{zzhao72, hwang339, jwu834, yyan55}@uic.edu
yuzhang.shang@ucf.edu, gaoliu@cisco.com

## Abstract

Dataset distillation aims to synthesize a small dataset from a large dataset, enabling the model trained on it to perform well on the original dataset. With the blooming of large language models and multimodal large language models, the importance of multimodal datasets, particularly image-text datasets, has grown significantly. However, existing multimodal dataset distillation methods are constrained by the Matching Training Trajectories algorithm, which significantly increases the computing resource requirement, and takes days to process the distillation. In this work, we introduce EDGE, a generative distillation method for efficient multimodal dataset distillation. Specifically, we identify two key challenges of distilling multimodal datasets with generative models: 1) The lack of correlation between generated images and captions. 2) The lack of diversity among generated samples. To address the aforementioned issues, we propose a novel generative model training workflow with a bi-directional contrastive loss and a diversity loss. Furthermore, we propose a caption synthesis strategy to further improve text-to-image retrieval performance by introducing more text information. Our method is evaluated on Flickr30K, COCO, and CC3M datasets, demonstrating superior performance and efficiency compared to existing approaches. Notably, our method achieves results $18\times$ faster than the state-of-the-art method. Our code will be made public at https://github.com/ichbill/EDGE.

## 1 Introduction

Dataset distillation [53] seeks to synthesize a compact dataset from a larger one, enabling a model trained on the distilled dataset to achieve strong performance on the original dataset. Typical kernel-based dataset distillation [53, 64] aims to match the performance of the synthetic dataset with the original dataset. The matching training trajectories (MTT) approaches [2, 19, 5, 9, 63, 12] have been proven effective for the distillation of small datasets. Some works [12, 63] even achieve a lossless performance from the original dataset. Recently, aiming the large-scale dataset, decoupled dataset distillation methods [45, 57], and generative distillation methods [44, 11, 49] have been proposed, which enhances scalability by improving the distillation efficiency.

With the emergence of large language models (LLM) and multimodal large language models (MLLM) [25, 23, 24, 16, 38], multimodal datasets become essential, especially image-text datasets. The first work on Vision Language Dataset Distillation (VLDD) [55] matches the expert training trajectories for both the image and text encoders and applies Low-Rank Adaptation [14] to reduce the computing requirements. The following work, LoRS [56], utilizes similarity mining during the distillation to improve the performance. However, these MTT-based methods require substantial time

---

*Corresponding Author.

39th Conference on Neural Information Processing Systems (NeurIPS 2025).

and computational resources to distill datasets. As illustrated in Figure 1, it takes about four days for MTT-VL to distill a dataset. The memory peak usage of MTT-VL reaches 200 GB for distilling 500 pairs and 320 GB for 100 pairs. For LORS [56], it takes about a week (150 hours) to distill only 500 image-text pairs. Therefore, time consumption becomes one of the main drawbacks of current VLDD methods.

Generative dataset distillation methods [11, 44] leverage large-scale pre-trained diffusion models [34] to distill image datasets for image classification tasks. However, such single-modality generative distillation methods are unsuitable for multimodal scenarios because they are designed for a limited number of classes and cannot be generalized to image-caption datasets where the captions are not constrained by a fixed number of categories. Additionally, directly applying multimodal generative models [34] to VLDD will encounter two main issues: 1) The lack of correlation between generated images and captions. We observe that directly applying generative models to distillation yields a suboptimal performance. This is mainly because the diffusion model training concentrates on sample-wise noise prediction instead of image-text correspondence, which is the most important aspect of the image-text contrastive learning (ITC) task on vision language datasets. 2) The lack of diversity among the generated samples. In dataset distillation, the synthetic dataset only contains less than 5% of the original data, so the generalization of the dataset is essential for the distilled dataset generation.

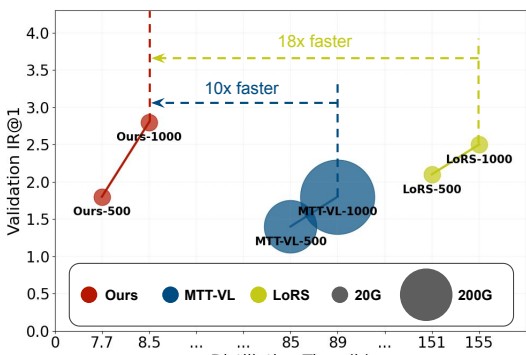

Figure 1: **Performance *vs*. resource usage.** Existing methods require substantial computation time and memory to distill datasets, whereas our method achieves up to $18\times$ faster processing and $16\times$ lower memory usage while delivering competitive performance. The experiments are conducted on NVIDIA RTX A5000 GPUs.

To address the aforementioned issues, in this work, we introduce **E**fficient Multimodal Dataset **D**istillation via **GE**nerative Models (EDGE), which utilizes generative priors for distilling vision-language datasets. Our approach incorporates a novel training workflow that integrates a bi-directional contrastive loss inspired by InfoNCE [31] and diversity loss inspired by Minimax [11] to improve image-text correlation and increase the diversity of distilled datasets. The EDGE framework begins by introducing noise to the image latent representation, predicting the noise with a conditioning embedding, and obtaining the denoised image latent. Using the denoised image latent and the corresponding text embedding, we apply the proposed losses as follows: 1) Contrastive loss supervises the generative model to produce content that is highly relevant to the given conditioning, aligning image and text representations effectively. 2) Diversity loss increases the variability of the generated samples by pushing apart the sample features, encouraging the synthetic dataset to reflect the same distribution as the original dataset. Additionally, to further improve performance in text-to-image retrieval tasks, we incorporate a caption synthesis strategy. This involves generating additional captions for images in the synthetic dataset using MLLMs, providing sufficient text information for the evaluation model training and boosting retrieval performance.

We evaluate our methods on multiple vision language datasets, including Flickr30K [33], COCO [22], and Conceptual Captions 3 Million (CC3M) [42]. Compared to existing VLDD methods, our method not only achieves more efficient distillation on Flickr30K and COCO, but also extends the capability to distill the large-scale CC3M dataset. As illustrated in Figure 1, our method significantly reduces the distillation time, achieving comparable performance to existing methods within just a few hours, compared to the days required by existing approaches. Furthermore, we successfully distill CC3M, a dataset containing approximately three million image-text pairs. Notably, most existing dataset condensation methods, including coreset selection, are infeasible for CC3M due to their extremely high computational resource requirements.

Our contributions are summarized as follows:

- We identify the efficiency issue of current MTT-based vision language dataset distillation methods and propose a generative vision language distillation method, EDGE, to efficiently distill the vision language datasets.

- We identify two key challenges in applying generative methods to VLDD, and introduce a novel training workflow incorporating contrastive loss and diversity loss. We further introduce caption synthesis to improve the text-to-image retrieval performance.
- We conduct empirical comparisons with existing methods, demonstrating competitive performance while reducing computational costs by 18 times compared to the SOTA approach.

## 2 Related works

### 2.1 Dataset Distillation

Dataset distillation (DD) [53] aims to compress a large dataset into a small synthetic dataset while preserving the performance of the models trained on it. Dataset distillation methods can be categorized into several classes: performance matching [53, 64, 27, 29, 28, 30, 3], parameter matching [61, 18, 15, 2, 5, 63, 12, 21, 50], distribution matching [60, 51, 39, 62, 36, 58, 7, 52, 6, 26, 20] and decoupled distillation methods [57, 40, 45, 41, 8, 59, 43, 48, 4]. Recently, generative distillation methods [11, 44, 49] synthesize images using generative models to efficiently distill large-scale datasets. Minimax Diffusion [11] introduces additional minimax criteria in the generative training to enhance the performance of the generated images of the diffusion model. D$^4$M [44] has explored the use of diffusion models to improve cross-architecture generalization and reduce computational overhead by constraining the consistency of the real and synthetic image spaces.

While generative distillation methods have proven successful for image classification tasks, existing single-modality approaches are not suitable for multimodal scenarios. These methods are designed for a fixed number of classes in both training and test datasets and cannot be generalized to image-caption datasets, where captions are not constrained to predefined categories.

### 2.2 Vision Language Dataset Distillation

While the majority of the dataset distillation community focuses on the image classification task in dataset distillation, recent works [55, 56] began to explore the new task of image-text contrastive learning (ITC) on vision language dataset. The first work of multimodal dataset distillation [55] matches both the image and text encoders and applies LoRA [14] to reduce the computing requirements. In particular, to overcome the challenge that the image-text contrastive learning dataset does not have discrete classes, this work first proposed jointly distilling image-text pairs in a contrastive manner. The following work, LoRS [56], utilizes similarity mining during distillation for performance improvement. Specifically, a similarity matrix for synthetic image-text pairs is calculated during data synthesis to explicitly describe the relevance of each image-text pair in the distilled dataset.

However, existing VLDD methods struggle with the distillation efficiency and the scale of the datasets, which is mainly due to the use of the Matching Training Trajectories (MTT) algorithm [2]. MTT obtains synthetic data through a bi-level optimization process, which involves an inner loop for model updates and an outer loop for synthetic data updates. Not only does the required large number of unrolled iterations during optimization cause immense computational costs, but generating expert trajectories itself is also extremely time-consuming. In this work, we propose EDGE to efficiently distill datasets via the generative method and scale up the large-scaled CC3M [42] dataset.

## 3 Methods

### 3.1 Problem Formulation

Given a large target dataset $\mathcal{T} = \{x_i, y_i\}_{i=1}^{|\mathcal{T}|}$, where $x_i$ denotes the training data and $y_i$ denotes the corresponding annotation, general dataset distillation method aims to synthesize a small dataset $\mathcal{S} = \{x_i, y_i\}_{i=1}^{|\mathcal{S}|}$, where $|\mathcal{S}| << \mathcal{T}$, so that the model trained on $\mathcal{S}$ has the optimal performance on the target dataset $\mathcal{T}$. Specifically for image-text contrastive learning (ITC), the target dataset consists of $|\mathcal{T}|$ image-text pairs. Note that in practice, in a dataset, an image can have multiple captions, and a caption can also have multiple corresponding images. We consider the size of the datasets as the number of image-text pairs since the number of image-text pairs determines the actual training time.

In Section 3.2, we begin by introducing the advantages of introducing diffusion model priors and highlighting the limitations of directly applying them to vision-language dataset distillation. Then we present the workflow of EDGE to overcome the limitations. Section 3.3 presents the proposed

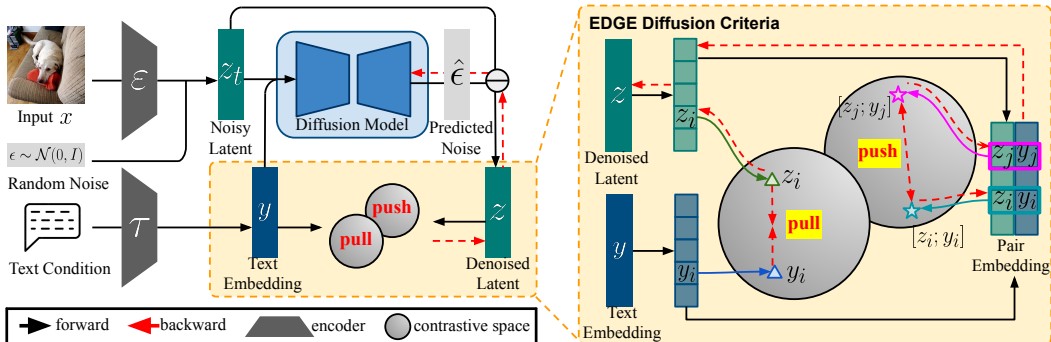

Figure 2: **The workflow of our method.** Given the input image $x$, text condition, we first get image latent $\mathbf{z}_0$ and text embedding $\mathbf{y}$ via the image encoder $\varepsilon$ and text encoder $\tau_\theta$. Random noise $\epsilon \sim \mathcal{N}(0, I)$ is added to the image latent, and the noise is then predicted using a U-Net [35] conditioned on $\mathbf{y}$. The denoised latent representation $\mathbf{z}$ is obtained as $\mathbf{z} = (\mathbf{z}_t - \sqrt{\beta^t} \cdot \hat{\epsilon})/\sqrt{\alpha^t}$, where $\hat{\epsilon}$ is the predicted noise. With the denoised latent $\mathbf{z}$ and text embedding $\mathbf{y}$, we employ a contrastive loss $\mathcal{L}_C$ to align corresponding image latents and text embeddings and a diversity loss $\mathcal{L}_D$ to encourage the diversity and generalization between concatenated image-text features.

contrastive loss and diversity loss. Finally, in Section 3.4, to further improve the text-to-image retrieval performance, we illustrate the caption synthesis process.

## 3.2 EDGE Diffusion for VLDD

Diffusion models learn the distribution of a dataset by progressively adding Gaussian noise to images and then reversing this process to reconstruct the original data. For instance, in the latent diffusion model (LDM) [34], the training process is described as follows for a given training image $x$. In the forward noising process, a vision encoder $\varepsilon$ encodes the image into the latent representation $\mathbf{z}_0 = \varepsilon(x)$. Gaussian noise $\epsilon \sim \mathcal{N}(0, I)$ is incrementally added to the initial latent code $z_0$, resulting in: $\mathbf{z}_t = \sqrt{\bar{\alpha}_t} \mathbf{z}_0 + \sqrt{1 - \bar{\alpha}_t} \epsilon$, where $\bar{\alpha}_t$ is a hyperparameter controlling the noise level at step $t$. Next, a decoder reconstructs this latent code back to the image space. With conditioning vector $\mathbf{y}$ encoding additional information, the diffusion model is trained to minimize the mean squared error between the predicted noise $\hat{\epsilon} = \epsilon_\theta(\mathbf{z}_t, \mathbf{y})$ and the added noise $\epsilon \sim \mathcal{N}(0, I)$:

$$\mathcal{L} = ||\epsilon_\theta(\mathbf{z}_t, \mathbf{y}) - \epsilon||_2^2, \tag{1}$$

where $\epsilon_\theta$ is a noise prediction network parameterized by $\theta$.

Diffusion models are proven efficient on dataset distillation methods for image classification tasks [11, 44]. However, when applying diffusion models to dataset distillation for ITC tasks, there are two main limitations to the distillation process: 1) The correspondence between image and text features is not fully explored. As mentioned above, the training of the diffusion model is sample-wise, which concentrates on noise prediction. Such a training scheme lacks the supervision of the correspondence of image and text features. 2) The lack of diversity among the synthetic samples. The number of samples is much smaller than the original datasets, so the diversity among the distilled samples is essential for dataset distillation.

To address the aforementioned issues, we propose the EDGE finetuning workflow as illustrated in Figure 2. Instead of calculating the mean squared error between the predicted noise $\epsilon_\theta(\mathbf{z}_t, \mathbf{c})$ and the ground truth $\epsilon$, we proposed two novel criteria for the vision language dataset distillation. Given the input image $x$, text condition, the encoders $\varepsilon$ and $\tau_\theta$ transform them into a compact latent representation, image latent $\mathbf{z}_0$ and text embedding $\mathbf{y}$, respectively. Then we add random noise $\epsilon \sim \mathcal{N}(0, I)$ to the image latent and predict the noise with U-Net [35] with the conditioning $\mathbf{y}$. With predicted noise $\hat{\epsilon}$, we use

$$\mathbf{z} = \frac{\mathbf{z}_t - \sqrt{\beta^t} \cdot \hat{\epsilon}}{\sqrt{\alpha^t}} \tag{2}$$

to get the de-noised latent $\mathbf{z}$, where $\alpha^t$ and $\beta^t$ are hyperparameters. The de-noised latent $\mathbf{z}$ and text embedding $\mathbf{y}$ are used to calculate the contrastive loss and the diversity loss, introduced in Section 3.3.

The contrastive loss is used to encourage the generative model to synthesize images that are highly relevant to the given text conditions, while the diversity loss is used to push away the image-text embeddings among different image-text pairs.

## 3.3 EDGE Diffusion Criteria

In this section, we begin by presenting the motivation behind the contrastive loss, followed by its design and impact. Next, we discuss the rationale for the diversity loss and provide its formulation. We then introduce the combined EDGE loss and conclude by explaining the process of generating the distilled dataset using our proposed method. **Contrastive Loss.** The goal of image-text contrastive learning (ITC) is to establish meaningful relationships between images and text. However, generative models, such as diffusion models, are typically trained to predict noise, without explicit supervision to capture image-text alignment. Instead of predicting the noise, we want to connect the latent space $z_i$ and $y_i$ for corresponding image $x_i$ and text $y_i$. Inspired by InfoNCE [31], we propose the contrastive loss that encourages aligned embeddings. Specifically, with the given image embedding $z$ and the text embedding $y$, the similarity score between normalized features $z_i$ and $y_i$ can be defined as $S_{ii} = \frac{z_i \cdot y_i}{\tau}$, where $\tau$ is the temperature parameter, which is set to 0.5 in experiments. Then we calculate the cross-entropy loss for both directions:

$$\mathcal{L}_{I \to T} = -\frac{1}{N} \sum_{i=1}^{N} \log \frac{\exp(S_{ii})}{\sum_{j=1}^{N} \exp(S_{ij})}, \mathcal{L}_{T \to I} = -\frac{1}{N} \sum_{i=1}^{N} \log \frac{\exp(S_{ii})}{\sum_{j=1}^{N} \exp(S_{ji})}, \tag{3}$$

where $S_{ij} = \frac{I_i \cdot T_j}{\tau}$ represents the similarity score between $z_i$ and $y_j$, and $S_{ji} = \frac{I_j \cdot T_i}{\tau}$ represents the similarity score between $z_j$ and $y_i$. With $\lambda$ controlling the weight ratio, the overall contrastive loss is defined as:

$$\mathcal{L}_{C} = \lambda \mathcal{L}_{I \to T} + \mathcal{L}_{T \to I}. \tag{4}$$

In equation 3, the loss encourages alignment between each image and its corresponding caption while also ensuring that each text matches the correct image. By incorporating this bi-directional contrastive objective, the diffusion model learns to capture the semantic relevance between images and text. Consequently, during sampling, the model generates image-text pairs with high relevance, which is beneficial for evaluation model training.

**Diversity Loss.** Compared to the original dataset, the distilled dataset distilled by generative methods [11] has limited diversity. This problem is even more evident in image-text contrastive (ITC) tasks compared to image classification, since the captions are not constrained to predefined categories. Diffusion model training is in a sample-wise manner and neglects this requirement for dataset distillation. To address this issue and ensure the distilled dataset accurately represents the original distribution, we focus on increasing the diversity of generated samples and enhancing the generalization of models trained on the synthetic dataset. To achieve this, we introduce a diversity loss that leverages the de-noised embedding $z$ and the text embedding $y$:

$$\mathcal{L}_{D} = \frac{2}{N(N-1)} \sum_{i=1}^{N} \sum_{j=i+1}^{N} \left( \frac{[z_i; y_i]}{\|[z_i; y_i]\|} \times \frac{[z_j; y_j]}{\|[z_j; y_j]\|} \right), \tag{5}$$

where $[z_i; y_i]$ represents the concatenation of the image and text embeddings $z_i$ and $y_i$. The term $\left( \frac{[z_i; y_i]}{\|[z_i; y_i]\|} \times \frac{[z_j; y_j]}{\|[z_j; y_j]\|} \right)$ indicates the similarity matrix of normalized concatenated embeddings. The diversity loss is designed to maximize the distance between these concatenated embeddings, effectively pushing them away in the embedding space. This encourages diversity in the representations of image-text pairs, promoting improved generalization and robustness.

Finally, by incorporating the contrastive loss $\mathcal{L}_C$ and the diversity loss $\mathcal{L}_D$ together, we formulate the EDGE loss as:

$$\mathcal{L}_{EDGE} = \mathcal{L}_{C} + \lambda \mathcal{L}_{D}. \tag{6}$$

The EDGE loss is used to train the diffusion model on the target dataset, to get the corresponding generative model. To synthesize the distilled dataset, we directly use diffusion model sampling for the required number of images. We first randomly select captions from the original dataset, and then use each caption as a condition to sample the image.

### 3.4 Caption Synthesis

We observed that when training a model on the synthetic dataset, it is often more challenging for the model to retrieve the correct image with a given text than to retrieve the correct text for a given image. As discussed in Section 1, applying the generative-based dataset distillation method to ITC tasks will counter the issue that the captions are not sufficient for the model training. To address this issue, we propose caption synthesis to generate more captions for each image. We developed a scalable approach to create as many captions as we want. Our method involves crafting specific prompt engineering templates that guide the Multimodal Large Language Models (MLLM) [25, 23, 24] to produce the captions for given images.

We start by gathering the images from the synthetic dataset. For each image, we consider a straightforward prompt template to generate captions effectively. As illustrated in Figure 3, we can use MLLM such as LLaVA [25] or GPT [32] model to generate any number of captions for a given image. The prompts we pass to different MLLMs are a little bit different. For LLaVA, we directly pass the prompt "*Describe the image in one sentence*" and we can get the required caption. GPT models tend to generate long sequences of text, and it often contains the phase that is used for interactions, such as "*the image contains*". Thus we change the prompt to "*Describe the image briefly in one sentence. Do not start with 'the image.'*"

During the model training stage, even though our method produces fewer images than captions, we consider the same image with different caption pairs as distinct image-text pairs. This is due to the similar computational resources required from the model training perspective.

Context

| LLaVA |
| --- |
| User => Describe the image in one sentence. |
| Assistant => A dog is laying on a couch with a red pillow on it. |

| GPT model |
| --- |
| User => Describe the image briefly in a short sentence. Do not start with 'the image'. |
| Assistant => A relaxed yellow labrador lies on a couch, resting its head on a bright orange toy. |

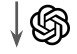

Augmented Caption

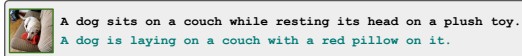

**A dog sits on a couch while resting its head on a plush toy. A dog is laying on a couch with a red pillow on it.**

Figure 3: **Illustration of the caption synthesis.** An image and a prompt are passed to a Multimodal Large Language Model (MLLM) to generate captions. Then the dataset contains $|\mathcal{S}|/\eta$ images with $|\mathcal{S}|$ captions, where $\eta$ is the caption per image (CPI) and each image is captioned by $\eta$ captions. Caption synthesis aims to provide sufficient text information for the model training.

## 4 Experiments

### 4.1 Datasets and Metrics

Following previous methods [55, 56], we evaluate our method on Flickr30k [33] and COCO [22] datasets for a fair comparison with existing methods. Flickr30K and COCO are image-text datasets with 31K and 123K images, where each image is paired with five captions. We also evaluated methods on a larger image-caption dataset, Conceptual Captions 3 Million (CC3M) [42], to validate the effectiveness of our method on large-scale datasets. CC3M contains about 3.3 million image-text pairs. The evaluation involves the metrics of top-K retrieval from both the image and the text perspectives. We denote the text-to-image retrieval as IR@K and image-to-text retrieval as TR@K.

### 4.2 Implementation Details

Following previous works [55, 56] on Vision Language Dataset Distillation, for the evaluation model, we adopt NFNet [1] as image encoder and BERT-base [17] as text encoder. The pre-trained weights are used, and the text encoder is frozen during evaluation. Our entire distillation process can be done on a single NVIDIA RTX A5000 GPU. For evaluation on the CC3M dataset, since the dataset is constructed with URLs and by the time we conduct experiments, some URLs are no longer available. Eventually, we collected 2.3M of 3.3M images for the CC3M training data. In addition, since the validation set is large, we first subsample a 1000 image-text pair validation subset and then evaluate the validation subset. In the distillation stage, following previous works [56], the images are resized to 224×224 resolution, and text embeddings are of 768 dimension.

### 4.3 Main Results

We compare our method with coreset selection methods [54, 10, 46], existing VLDD methods [55, 56] and a pre-trained generative model, Stable-diffusion v1.5 [34] on Flickr30K and COCO datasets.

Table 1: **Results of 500 pairs on Flickr30K and COCO dataset.** For Flickr30K, the dataset condensation ratio is 1.7%. For COCO, the condensation ratio is 4.4‰. **Bold** values indicate the best results, while underlined values denote the second-best results.

| Dataset | Ratio | Metric | Coreset Selection | | | | Dataset Distillation | | | | |
| --- | --- | --- | --- | --- | --- | --- | --- | --- | --- | --- | --- |
| | | | Rand | HERD | K-Cent | Forget | MTT-VL | TESLA | LoRS | SD | **EDGE** |
| Flickr30K | 1.7% | IR@1 | 2.4 | 3.0 | 3.5 | 1.8 | 6.6 | 1.1 | **10.0** | 3.1 | 6.7 |
| | | IR@5 | 10.5 | 10.0 | 10.4 | 9.0 | 20.2 | 7.3 | **28.9** | 11.5 | 21.0 |
| | | IR@10 | 17.4 | 17.0 | 17.3 | 15.9 | 30.0 | 12.6 | **41.6** | 18.5 | 30.5 |
| | | TR@1 | 5.2 | 5.1 | 4.9 | 3.6 | 13.3 | 5.1 | **15.5** | 4.6 | 13.3 |
| | | TR@5 | 18.3 | 16.4 | 16.4 | 12.3 | 32.8 | 15.3 | **39.8** | 15.1 | 35.6 |
| | | TR@10 | 25.7 | 24.3 | 23.3 | 19.3 | 46.8 | 23.8 | **53.7** | 22.2 | 47.5 |
| COCO | 4.4‰ | IR@1 | 1.1 | 1.7 | 1.1 | 0.8 | 1.4 | 0.8 | **2.1** | 1.2 | 1.8 |
| | | IR@5 | 5.0 | 5.3 | 6.3 | 5.8 | 5.1 | 3.6 | **8.0** | 5.1 | 6.5 |
| | | IR@10 | 8.7 | 9.9 | 10.5 | 8.2 | 8.9 | 6.9 | **13.7** | 9.0 | 11.2 |
| | | TR@1 | 1.9 | 1.9 | 2.5 | 2.1 | 2.3 | 1.7 | 2.8 | 2.2 | **2.9** |
| | | TR@5 | 7.5 | 7.8 | 8.7 | 8.2 | 8.4 | 5.9 | **9.9** | 8.7 | 9.5 |
| | | TR@10 | 12.5 | 13.7 | 14.3 | 13.0 | 13.8 | 10.2 | **16.2** | 14.9 | 15.7 |

Table 2: **Results of 1000 pairs on Flickr30K and COCO dataset. Bold** values indicate the best results, while underlined values denote the second-best results.

| Dataset | Ratio | Metric | Rand | LoRS | SD | **EDGE** |
| --- | --- | --- | --- | --- | --- | --- |
| Flickr30K | 3.4% | IR@1 | 3.9 | **11.2** | 3.3 | 9.9 |
| | | IR@5 | 13.0 | **31.3** | 11.9 | 28.2 |
| | | IR@10 | 20.7 | **42.9** | 19.1 | 40.5 |
| | | TR@1 | 5.3 | 14.4 | 5.0 | **14.5** |
| | | TR@5 | 17.1 | 37.5 | 15.9 | **38.3** |
| | | TR@10 | 27.7 | 50.5 | 24.8 | **51.7** |
| COCO | 8.8‰ | IR@1 | 1.6 | 2.5 | 1.8 | **2.8** |
| | | IR@5 | 6.5 | 9.4 | 6.9 | **9.8** |
| | | IR@10 | 11.5 | 15.4 | 11.9 | **16.2** |
| | | TR@1 | 2.4 | 3.6 | 2.8 | **3.9** |
| | | TR@5 | 9.9 | 11.8 | 10.2 | **13.0** |
| | | TR@10 | 16.7 | 19.1 | 16.8 | **21.0** |

Table 3: **Results on CC3M.** We compare our method with random selection and a pre-trained Stable Diffusion v1.5 model. The synthetic dataset contains 1000 image-text-pairs, which is only 0.3‰ of the original dataset. Other existing methods are not applicable due to long computation time or high memory requirements.

| Methods | IR@1 | IR@5 | IR@10 | TR@1 | TR@5 | TR@10 |
| --- | --- | --- | --- | --- | --- | --- |
| Rand | 0.1 | 0.4 | 0.9 | 0.0 | 0.3 | 0.9 |
| SD | 0.1 | 0.4 | 0.9 | 0.0 | 0.3 | 0.9 |
| **EDGE** | **0.2** | **0.5** | **1.0** | **0.1** | **0.7** | **1.1** |

Table 4: **CLIP score.**

| Methods | Flickr-30K | COCO | CC3M |
| --- | --- | --- | --- |
| MTT-VL | 0.2966 | 0.2264 | - |
| LoRS | 0.0077 | 0.0105 | - |
| **Ours** | **0.3227** | **0.3178** | **0.3086** |

Table 5: **FID score.**

| Methods | Flickr-30K | COCO |
| --- | --- | --- |
| MTT-VL | 210.0 | 276.3 |
| **Ours** | **88.1** | **83.1** |

Due to extremely long computing time, coreset selection methods [54, 10, 46] are hard to be applied on a large number of pairs. As for MTT-based DD methods [55, 5], distilling to a large number of pairs will lead to extremely high GPU memory usage.

**Flickr30K results.** We evaluate our method on the Flickr30K dataset and compare it with existing approaches. The distilled dataset contains 500 and 1000 image-text pairs, representing 1.7% and 3.4% of the original dataset, respectively. As shown in Table 1 and Table 2, our method achieves performance comparable to or even outperforms that of time-intensive SOTA methods across several metrics. Notably, compared to the baseline—a pre-trained Stable Diffusion v1.5 model—our approach demonstrates significant improvements, highlighting the effectiveness of the proposed techniques.

**COCO results.** We also evaluate our method on the COCO dataset in comparison with existing methods in Table 1 and Table 2. The distilled dataset contains 500 and 1000 image-text pairs, representing 4.4‰ and 8.8‰ of the original dataset, respectively. With 500 pairs, it achieves competitive performance relative to SOTA methods, which require a significantly longer distillation time. When distilling 1000 image-text pairs, our method shows an obvious advantage from both performance and efficiency perspectives. Similar to the Flickr30K results, our method substantially improves over the baseline pre-trained Stable Diffusion model, further validating its effectiveness.

**CC3M results.** For larger datasets like Conceptual Captions 3M (CC3M) [42], existing dataset distillation methods are not applicable due to the high resource demands of trajectory matching algorithms. Even coreset selection methods are difficult to apply, as their time consumption and memory usage scale rapidly. In contrast, our method is capable of distilling the CC3M dataset efficiently using only a single GPU. The distilled dataset contains 500 image-text pairs, representing 0.3‰ of the original dataset. We compare our approach with the random selection method and the pre-trained Stable Diffusion v1.5 model in Table 3. Our method outperforms the baseline methods, demonstrating the effectiveness of our method on CC3M dataset.

**More aspects of evaluation.** Furthermore, we provide the CLIP scores on Flickr30K, COCO, and CC3M, comparing with existing dataset distillation methods in Table 4. Table 5 assesses image

Table 6: **Comparison of required computing resources.** We provide the GPU hours required for different method to distill different datasets. "OOM" in the table indicates the method is not applicable due to extremely high memory usage.

| Dataset | Flickr30K | | | COCO | | | CC3M | | |
| # Pairs | 500 | 1000 | 1500 | 500 | 1000 | 1500 | 500 | 1000 | 1500 |
|---|---|---|---|---|---|---|---|---|---|
| MTT-VL [55] | 77.8 | 109.1 | 141.9 | 162.3 | 194.1 | OOM | OOM | OOM | OOM |
| LoRS [56] | 54.2 | 57.6 | 59.0 | 151.6 | 155.6 | 172.5 | OOM | OOM | OOM |
| **EDGE (ours)** | **13.6** | **14.3** | **14.9** | **7.7** | **8.5** | **9.2** | **31.7** | **32.5** | **33.2** |

Table 7: **Cross architecture evaluation.** For baseline method, we use NFNet+BERT to distill, and evaluate with various architectures. Compared to matching training trajectory based approaches, the dataset generated by our method has a better generalization.

| # Pairs | Ratio | Method | Evaluation model | COCO | | | | | |
| | | | | IR@1 | IR@5 | IR@10 | TR@1 | TR@5 | TR@10 |
|---|---|---|---|---|---|---|---|---|---|
| 1000 | 8.8‰ | LoRS | NfNet+Bert | 2.5 | 9.4 | 15.4 | 3.6 | 11.8 | 19.1 |
| | | | ResNet+Bert | 0.1 | 0.6 | 1.1 | 0.4 | 1.6 | 2.8 |
| | | | RegNet+Bert | 0.1 | 0.5 | 0.9 | 0.1 | 0.5 | 0.9 |
| 1000 | 8.8‰ | EDGE | NfNet+Bert | 2.8 | 9.8 | 16.2 | 3.9 | 13.0 | 21.0 |
| | | | ResNet+Bert | 2.5 | 9.5 | 16.0 | 3.7 | 12.7 | 20.1 |
| | | | RegNet+Bert | 2.3 | 8.4 | 14.1 | 2.6 | 10.7 | 17.7 |

quality using FID. Results show our method improves both text-image alignment and image quality, where the former is more crucial for retrieval tasks.

**Computational efficiency.** We provide the computing efficiency comparison of our method and the SOTA methods [55, 56]. As depicted in Table 6, for Flickr30K, our method is approximately 9 times faster than MTT-VL, and 4 times faster than LoRS. For the COCO dataset, our method is 22 times faster than MTT-VL and 18 times faster than LoRS. Not even mentioning that both existing methods [55, 56] are not applicable for large dataset CC3M [42]. For CC3M, our method is the only applicable dataset distillation method, and our method is able to distill the dataset in an efficient manner. The GPU hour usage is even less than the baseline methods on the Flickr30K dataset.

**Cross-architecture experiments.** In this work, we also evaluate the cross-architecture generalization of our method. As shown in Table 7, we evaluate performance by changing the image encoder, and our method significantly outperforms the baseline. The results highlight that the cross-architecture performance of the baseline method [56] is highly dependent on the model used during distillation. For example, when synthetic datasets are distilled with NFNet and BERT, evaluation using the exact same model architectures yields strong performance. However, substituting NFNet with ResNet [13] or RegNet [37] causes a significant performance drop in both image and text retrieval tasks. In contrast, our method, which is not related to a specific model architecture during training, maintains consistent performance across various model architectures.

## 4.4 Ablation Study

**Ablation on different components.** Table 8 states the effectiveness of each component of our method. The experiments are performed on the COCO [22] dataset, distilling into 500 image-text pairs. The baseline "SD v1.5" in the table indicates the baseline model, a pretrained stable-diffusion model v1.5 [34] used in our method. "+ $\mathcal{L}_C$" and "+ $\mathcal{L}_D$" indicates the generative model fine-tuned with our proposed workflow in Figure 2 and corresponding losses.

The pre-trained stable diffusion model shows low performance on the evaluation dataset, indicating that the vanilla generative models, which are not designed for dataset distillation tasks, do not guarantee the correspondence for image-text pairs. By adding our proposed fine-tuning workflow and the corresponding losses, both IR and TR have been improved. By further adding the post-training caption synthesis, the performance of image-text retrieval is further improved, indicating the effectiveness of proposed methods.

**Effect of different diffusion models.** The diffusion model used to distill the dataset is an important aspect of our method. We evaluate two widely used diffusion models: Stable Diffusion v1.5 (denoted as "SD v1.5" in the table) and Stable Diffusion 3 (denoted as "SD 3") in this work. We compare two models from both perspectives of performance and efficiency. The performance of the methods is depicted in Table 9, where the performance of two diffusion models is similar. However, the sampling

Table 8: **Effect of different components.** The experiments are conducted on COCO dataset. The synthetic dataset contains 500 image-text pairs. The evaluation model is NFNet+BERT. "CS" denotes the Caption Synthesis strategy.

| Methods | IR@1 | IR@5 | IR@10 | TR@1 | TR@5 | TR@10 |
|---|---|---|---|---|---|---|
| SD v1.5 | 1.2 | 5.1 | 9.0 | 2.2 | 8.7 | 14.9 |
| + $\mathcal{L}_C$ | 1.3 | 5.1 | 9.2 | 2.3 | 8.3 | 13.7 |
| + $\mathcal{L}_C$ + $\mathcal{L}_D$ | 1.4 | 5.6 | 10.1 | 2.3 | 8.9 | 14.4 |
| + $\mathcal{L}_{EDGE}$ + CS | 1.8 | 6.5 | 11.2 | 2.9 | 9.5 | 15.7 |

Table 9: **Comparison of different generative models.** SD represents Stable Diffusion. Stable Diffusion v1.5 takes about 0.4 hour to distill 500 images, while it takes approximately 1.6 hours to distill 500 images for Stable Diffusion 3.

| Methods | IR@1 | IR@5 | IR@10 | TR@1 | TR@5 | TR@10 |
|---|---|---|---|---|---|---|
| SD v1.5 | 1.2 | 5.1 | 9.0 | 2.2 | 8.7 | 14.9 |
| SD 3 | 1.2 | 5.2 | 9.4 | 2.4 | 8.6 | 14.7 |

Table 10: **Ablation study on caption per image (CPI).** On COCO dataset, when CPI=2, the model achieves the best performance.

| Methods | IR@1 | IR@5 | IR@10 | TR@1 | TR@5 | TR@10 |
|---|---|---|---|---|---|---|
| CPI = 1 | 1.9 | 6.9 | 11.8 | 2.7 | 9.7 | 16.0 |
| CPI = 2 | **2.8** | **9.8** | **16.2** | **3.9** | **13.0** | **21.0** |
| CPI = 5 | 1.2 | 4.7 | 8.2 | 2.0 | 7.0 | 12.3 |

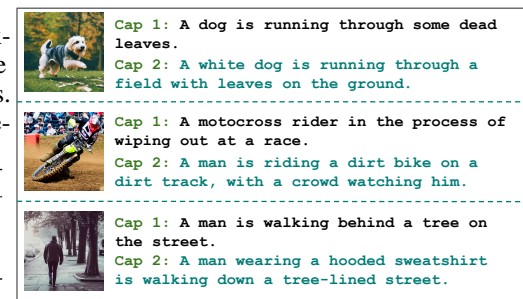

Cap 1: A dog is running through some dead leaves.
Cap 2: A white dog is running through a field with leaves on the ground.

Cap 1: A motocross rider in the process of wiping out at a race.
Cap 2: A man is riding a dirt bike on a dirt track, with a crowd watching him.

Cap 1: A man is walking behind a tree on the street.
Cap 2: A man wearing a hooded sweatshirt is walking down a tree-lined street.

Cap 1: A sports mascot during a performance.
Cap 2: A mascot in a yellow jacket and a black shirt is standing on a field.

Figure 4: **Examples of images and captions of the distilled dataset.** For each example image, two corresponding captions are provided. The **green** captions are generated by caption synthesis.

Table 11: **Comparison of different MLLMs.** Post-training strategies perform better than pre-processing strategies. With captions by LLaVA, the model achieves the best performance.

| Methods | IR@1 | IR@5 | IR@10 | TR@1 | TR@5 | TR@10 |
|---|---|---|---|---|---|---|
| LLaVA | **2.8** | **9.8** | **16.2** | **3.9** | **13.0** | **21.0** |
| GPT-4o-mini | 2.5 | 8.8 | 14.9 | 3.7 | 12.5 | 20.0 |
| Llama | 1.2 | 5.0 | 8.8 | 2.4 | 8.5 | 13.6 |

time of stable diffusion 3 is four times longer than that of Stable Diffusion v1.5. Consequently, we selected Stable Diffusion v1.5 as the primary model for our results.

**Effect of caption per image.** For caption synthesis, the number of captions per image (CPI) is essential. To investigate its impact on the model's performance, we vary the CPI during caption synthesis while keeping the total number of image-text pairs constant. The evaluation is done on COCO dataset, distilling the dataset into 500 image-text pairs. For CPI=1, CPI=2, CPI=5, we sample 500, 250, 100 images.

There are two main observations from Table 10: 1) Compared to the "vanilla" synthetic dataset, where each image is paired with only one caption, generating additional captions assists with the evaluation model training. This observation is consistent with the motivation of our caption synthesis, harm the performance of the evaluation model. This occurs because, with a fixed number of image-text pairs, increasing the CPI leads to the reduction of the total number of unique images. Consequently, the reduced number of images becomes insufficient for effective model training.

**Effect of different MLLMs.** The MLLM used for caption synthesis is also essential in our method. In this work, we compare three MLLM/LLM models with two different strategies. The first strategy, pre-processing, involves rephrasing captions before image generation. The second strategy, post-training caption synthesis, involves rephrasing captions after image generation, as illustrated in Section 3.4. The models evaluated in our study include LLava [25], GPT-4o-mini [32], and Llama [47]. For LLava, we adopt the post-training strategy, while for Llama, we use the pre-processing strategy. For the GPT model, both pre-processing and post-training strategies are applicable.

As illustrated in Table 11, the pre-processing strategy, regardless of the model used for rephrasing captions, proves to be less effective compared to post-training strategies. For the post-training methods, captions generated by the LLaVA model are more beneficial for training the evaluation model than those generated by the GPT model. Therefore, we adopt the post-training caption synthesis strategy with the LLaVA model as our primary approach.

## 4.5   Dataset Visualization

We visualized the images with corresponding captions generated by our method in Figure 4. Compared to previous VLDD methods [55, 56], the images generated by our method exhibit a realistic, high-quality appearance. The **green** captions are generated by our caption synthesis. The captions exhibited

in the figure are generated by LLaVA [25]. We can observe that compared to the image caption from the original dataset, the synthesized caption tends to capture more features from the image.

# 5 Conclusion

In this work, we identify key limitations in current vision language dataset distillation methods, including poor efficiency and limited scalability. To address these issues, we propose EDGE, a generative vision language dataset distillation approach designed to efficiently distill large-scale multimodal datasets. Specifically, we introduce a novel training workflow for generative models tailored to the image-text dataset distillation task. Additionally, we implement a post-training caption synthesis step to further enhance performance. Our method not only outperforms existing approaches on small datasets but also enables effective distillation of large datasets.

**Acknowledgments:** This research is supported by NSF IIS-2525840, CNS-2432534, ECCS-2514574, NIH 1RF1MH133764-01 and Cisco Research unrestricted gift. This article solely reflects opinions and conclusions of authors and not funding agencies.

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

## A    Training details

We provide the detailed hyperparameter settings used during diffusion model training. Following the approach in [11], we fine-tune only a small subset of the diffusion model's parameters. The weight $\lambda$ in Equation 6 is set to 1. For Flickr30K dataset, the learning rate is set to $1 \times 10^{-4}$, with a batch size of 8 and a total of 16 training epochs. For COCO dataset, the learning rate is set to $1 \times 10^{-4}$, with a batch size of 8 and a total of 8 training epochs.

## B    Qualitative results

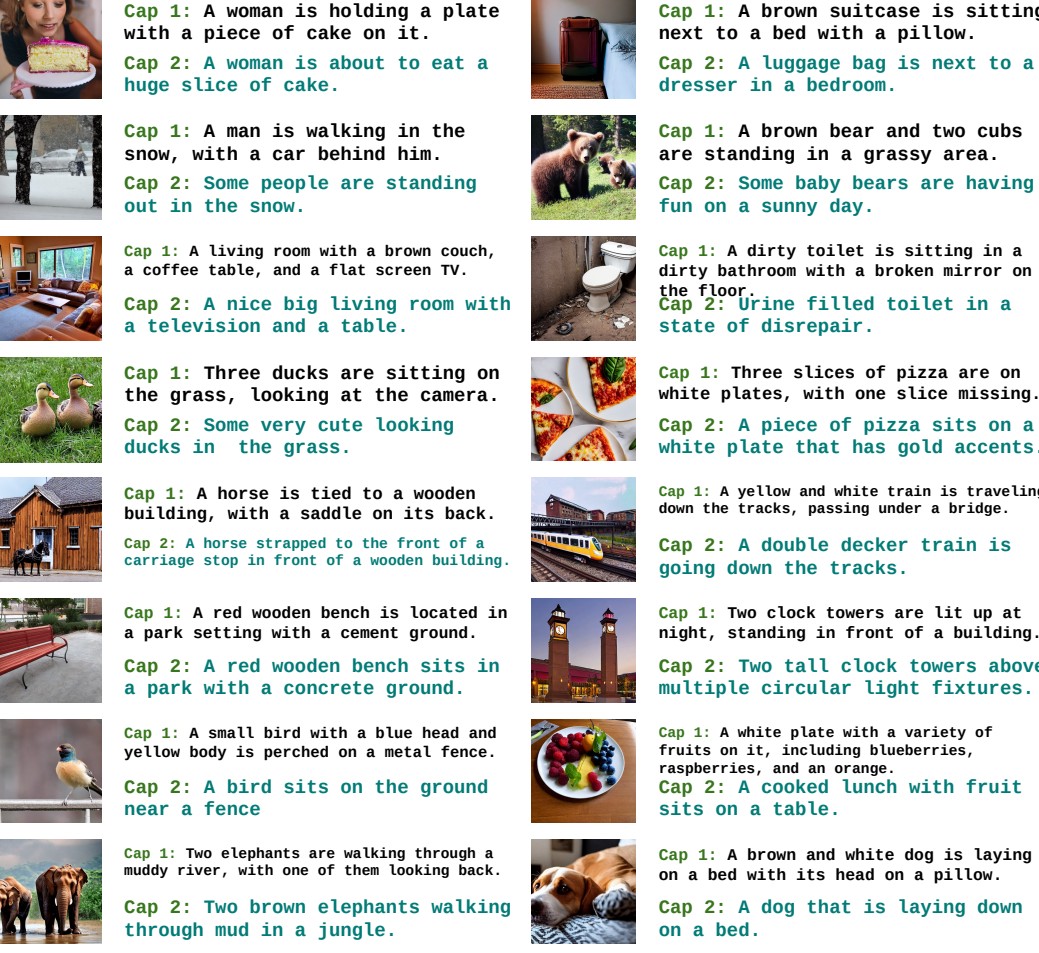

Figure 5: More qualitative results of the distilled dataset.

## C    Broader Impacts and Limitations

The societal impacts of our work are multifaceted. Positively, it contributes to greater resource efficiency, accelerates advancements in artificial intelligence, improves accessibility, and can enhance data privacy. However, it also introduces potential drawbacks, such as the possibility of information loss, the introduction of security vulnerabilities, and the complication of intellectual property rights. A careful balance between these advantages and challenges is paramount to ensure the responsible and ethical application of this technology.

We recognize that the current approach may exhibit limitations when applied to highly specialized domains such as medical imaging, where the data distribution, structural characteristics, and annotation

protocols differ substantially from those of general-purpose datasets used in our experiments. These domain-specific complexities may not be fully captured by the existing framework. Future research could address this limitation by incorporating domain-informed priors, adapting the optimization objectives, or designing customized representations that better align with the underlying structure and semantics of such specialized data.

