# OpenReview forum: "Efficient Multimodal Dataset Distillation via Generative Models"
_NeurIPS.cc/2025/Conference — NeurIPS 2025 poster_

### Official Review · Reviewer_r1pa · 2025-06-23

**Clarity:** 3
**Significance:** 3
**Originality:** 4
**Rating:** 5
**Confidence:** 2

**Summary:**

The paper introduces a novel generative method (EDGE) for multimodal dataset distillation, addressing the limitations of existing approaches that are computationally intensive and lack scalability for large image-text datasets. The gist of the method is in the reformulation of the contrastive loss inspired by InfoNCE that alignes text-image embeddings, and added diversity loss. The authors show that they can match or outperform existing methods at a fraction of computational cost, including distilling large datasets that were out of reach for previous methods.

**Questions:**

* One of the 2 objectives is to increase the diversity of the distilled dataset - while that indeed seems to help as demonstrated with Table 8, do you have any other evals that could further substantiate the claims about increased diversity of the dataset? (eg. using Vendi score, DPP or some other diversity metric)
   * Also the same question for text-image alignment evals (have you considered using TIFA/VNLI/Gecko/DSG?)

* Do you have any intuition about why are CLIP scores for LORS so much lower compared to EDGE? This seems particularly surprising given that performance is on-par.  Similarly, FID scores are very high, would be good to have some intuition on why (also why did you not include LoRS results here?)

* Limitations: Are there any pathological corner cases for your method that may have not been captured with the datasets you're looking at?

**Ethical Concerns:**

["NO or VERY MINOR ethics concerns only"]

**Final Justification:**

The authors adequately addressed my questions and have provided further experimental data that solidifies the existing results. I am happy to maintain the existing recommendation.

**Limitations:**

yes

**Paper Formatting Concerns:**

Tables could use a bit more formatting (eg.. page 9 seems pretty crammed)

**Quality:**

4

**Strengths And Weaknesses:**

Strengths:
* Strong results substantiated with multiple experiments, the proposed method achieves results comparable to related work (LoRS) yet at a much lower computational footprint to the extent that it makes it possible to distill very large datasets (such as CC3M)
* Novel proposal for generative loss
* Comprehensive ablation studies (different losses, different MLLMs) to demonstrate robustness
* The paper is very transparent about methods/datasets it uses, including highlighting where the competing methods perform better

Weaknesses:
* Evals could be enriched by adding a few more different methods to evaluate text-image alignment and diversity (for example, by considering VQA-based text-to-image alignment metrics)
* A minor point, but it would be nice to gain some intuition about how are distilled datasets different/similar between EDGE and other methods (eg examples that would substantiate "images generated by our method exhibit a realistic, high-quality appearance")
* No clear discussion of limitations of your method
* A bit more clarity on the scale of improvements would be fair: For example in Table 3: "our method outperforms the baseline methods" --> That "outperform" here carries a lot of weight here. The numbers are still very close to random baseline and in total those are very small numbers so I would advise to reframe this sentence to highlight that distilling CC3M is indeed still an open challenge.

---

> ### Author Rebuttal · Authors · 2025-07-30
>
> We sincerely thank you for the valuable questions and comments. For the concerns and questions, here are our responses:
> ### **W1: Evals could be enriched by adding a few more different methods to evaluate text-image alignment and diversity.**
>
> **Table 1.** Feature analysis compared to SOTA methods.
>
> | Methods           | MMD ↓   | NNO ↑   | Entropy ↑ |
> |-------------------|---------|---------|------------|
> | MTT-VL            | 0.0293  | 71.04%  | 7.79       |
> | LoRS              | 0.0261  | 78.11%  | 7.87       |
> | Stable-diffusion  | 0.0286  | 70.55%  | 8.01       |
> | EDGE (ours)       | 0.0242  | 79.68%  | 8.34       |
>
> **Table 2.** VQA metrics evaluation.
>
> | Methods         | Tifa-score ↑ | DSG score with dependency ↑ | DSG score without dependency ↑ |
> |----------------|------------|----------------------------|------------------------------|
> | LoRS           | 0.856      | 0.852                      | 0.876                        |
> | Stable-diffusion | 0.824    | 0.771                      | 0.805                        |
> | EDGE (ours)    | 0.877      | 0.874                      | 0.894                        |
>
> Thank you for your suggestions. To analyze dataset diversity, we now report Mean Distribution Discrepancy (MDD), Nearest Neighbor Overlap (NNO), and Entropy in Table 1, alongside the FID and CLIP‑based scores already in the paper. In the revised manuscript, we have also incorporated TIFA [1] and DSG [2], as shown in Table 2. These additional metrics consistently confirm that the distilled dataset maintains strong semantic alignment while exhibiting higher visual and textual variety than competing approaches. We will place the full numerical results and implementation details in the appendix and reference them in the main text to make these contributions clear to readers and reviewers.
>
> [1] TIFA: Accurate and Interpretable Text-to-Image Faithfulness Evaluation with Question Answering. ICCV 2023.
>
> [2] Davidsonian Scene Graph: Improving Reliability in Fine-grained Evaluation for Text-to-Image Generation. ICLR 2024.
>
> ### **W2: A minor point, but it would be nice to gain some intuition about how are distilled datasets different/similar between EDGE and other methods.**
>
> Thank you for your suggestions. In the revised version, we will include a concise qualitative comparison that places representative samples from EDGE next to those generated by MTT‑VL and LoRS. As the NeurIPS rebuttal format prohibits the inclusion of new figures, we summarize the main observations here. In addition, both MTT-VL and LoRS also provide qualitative examples in their paper. Compared to these baselines, EDGE produces images with sharper edges and more faithfully preserved fine details such as fur, foliage, and specular highlights, whereas baseline outputs often exhibit noticeable blur and noise. Quantitative results corroborate the visual differences: EDGE attains substantially lower FID scores, indicating closer alignment with the original data distribution and higher perceptual quality.
>
> ### **W3 & Q3: No clear discussion of limitations of your method. Limitations: Are there any pathological corner cases for your method that may have not been captured with the datasets you're looking at?**
>
> Thank you for your thoughtful feedback. We acknowledge that our current method may have limitations when applied to highly specialized domains such as medical imaging, where data characteristics can differ significantly from the general datasets used in our evaluation. These domains may involve unique structural patterns or annotation constraints that are not fully captured in our current experiments. We believe future work could explore adapting our approach to such specialized settings by incorporating domain-specific priors or tailoring the optimization process accordingly. We will add a detailed discussion of these limitations in the revised manuscript.
>
> ### **W4: Revise the statement of the scale of improvements.**
> Thank you for the constructive feedback. We appreciate your observation regarding the phrasing in Table 3 and fully agree that clarity in interpreting the scale of improvements is important. In our revision, we will adjust the wording to more accurately reflect the relative improvements and acknowledge the overall difficulty of the task. Rather than stating that our method "outperforms" the baselines in absolute terms, we will highlight that the performance gains, though modest in magnitude, consistently demonstrate the advantages of our distillation approach over existing methods. We will also explicitly note that distilling CC3M remains a challenging open problem, which our method takes a step toward addressing. This reframing should provide a more balanced and accurate interpretation of the results.
>
> ### **Q1: Evals that could further substantiate the claims about increased diversity of the dataset.**
>
> Thank you for the thoughtful comment. In addition to Table 8, we have further evaluated the diversity of the distilled dataset using three complementary metrics: Minimum Distance Distribution (MDD), Nearest Neighbor Overlap (NNO), and Entropy. MDD quantifies the compactness of sample representations, NNO reflects neighborhood uniqueness, and Entropy measures the overall uncertainty in token usage. As shown in the table below, our method EDGE consistently achieves lower MDD, higher NNO, and higher Entropy compared to other baselines, indicating that EDGE generates more diverse and less redundant samples. These results provide additional evidence supporting our claim that enhancing diversity contributes to the improved performance of our distilled dataset. We appreciate the suggestion of using Vendi score or DPP and will consider including those in future work to further strengthen the analysis.
>
> ### **Q1.1: Text-image alignment evals.**
>
> Thank you for the valuable suggestion. We have now evaluated our method using the TIFA benchmark to better assess text-image alignment performance. As shown in the table below, EDGE achieves a TIFA score of 0.8773, outperforming both LoRS and Stable Diffusion. Additionally, we have evaluated our method using DSG scores, both with and without dependency annotations. EDGE again demonstrates superior performance, achieving scores of 0.874 and 0.894, respectively. These results reinforce the strength of EDGE in aligning image and text modalities. We appreciate the reviewer’s insight and will consider including additional benchmarks such as VNLI and Gecko in future work to provide a more comprehensive evaluation.
>
> ### **Q2: Do you have any intuition about why are CLIP scores for LORS so much lower compared to EDGE? This seems particularly surprising given that performance is on-par. Similarly, FID scores are very high, would be good to have some intuition on why (also why did you not include LoRS results here?)**
>
> **Table 3.** FID score comparison.
>
> | Methods      | Flickr-30K | COCO  |
> |--------------|------------|-------|
> | MTT-VL       | 210.0      | 276.3 |
> | LoRS         | 114.7      | 91.6  |
> | EDGE (ours)  | 88.1       | 83.1  |
>
> Thank you for the thoughtful comment. Regarding the FID scores, we believe the notably higher FID of MTT-VL and LoRS compared to EDGE stems from fundamental differences in how the datasets are constructed. MTT-VL and LoRS rely on gradient-based updates to synthetic images guided by trajectory matching loss, which can lead to artifacts or unrealistic samples. In contrast, EDGE first learns a compact representation of the target dataset and then generates images using the fine-tuned generative model conditioned on this representation. This design allows EDGE to produce more visually realistic samples, resulting in significantly lower FID scores. We included the FID scores for LoRS in Table 3.
>
> As for the lower CLIP scores observed for LoR, we believe this arises from how LoRS updates text embeddings. Specifically, LoRS perturbs the original text embeddings by adding gradients from the trajectory matching loss. While this may help align synthetic image-text pairs for training, it can disrupt the semantic structure of the original text embeddings, which negatively impacts CLIP scores that depend on the global alignment between image and text representations. Nevertheless, LoRS maintains a learned image-text similarity matrix that guides evaluation model training, compensating for this distortion and enabling it to achieve reasonable performance despite the lower CLIP alignment.

---

> ### Comment · Reviewer_r1pa · 2025-08-04
>
> Thank you for addressing the concerns and providing additional experiments that support the claims made in the paper. I am satisfied with the responses and will maintain my rating.

---

> > ### Author Response · Authors · 2025-08-04
> >
> > Thank you for your thoughtful review and for taking your time to reconsider our responses. We truly appreciate your engagement and are glad to hear that our clarifications addressed your concerns.
> >
> > Best regards,
> >
> > The authors

---

### Official Review · Reviewer_3o9X · 2025-07-01

**Clarity:** 4
**Significance:** 4
**Originality:** 4
**Rating:** 5
**Confidence:** 4

**Summary:**

This paper proposes EDGE, an efficient generative method for multimodal dataset distillation, aiming to address the high computational costs and long distillation time of existing MTT-based methods. It identifies two key challenges in distilling multimodal datasets with generative models: the lack of correlation between generated images and captions, and the lack of diversity among generated samples. To tackle these, EDGE introduces a novel training workflow with a bi-directional contrastive loss and a diversity loss, as well as a caption synthesis strategy to improve text-to-image retrieval performance. Evaluations on Flickr30K, COCO, and CC3M datasets show that EDGE achieves superior performance and efficiency, being up to 18× faster than the state-of-the-art method while distilling large-scale datasets effectively.

**Questions:**

1. Why is the CC3M dataset particularly challenging for existing dataset distillation methods, and how does EDGE address this?
2. What is the role of the diversity loss in EDGE, and how does it differ from traditional dataset distillation approaches?
3. What is the purpose of the caption synthesis strategy in EDGE, and how does it work?
4. What is the role of the diversity loss in EDGE, and how does it differ from traditional dataset distillation approaches?

**Ethical Concerns:**

["NO or VERY MINOR ethics concerns only"]

**Limitations:**

This paper has two limitations. It relies on pre-trained generative models like Stable Diffusion, which may introduce biases and limit generalizability to specialized domains. The diversity loss, while effective empirically, lacks a rigorous theoretical foundation for matching the original dataset’s distribution.

**Paper Formatting Concerns:**

The References section includes proper citations (e.g., [1], [2]), but some entries (e.g., ) have incomplete titles or missing details (e.g., "Llava-next: Improved reasoning, ocr, and world knowledge, January 2024."). Additionally, the reference list may contain duplicate entries (e.g., [2] and [3] both reference "Dataset distillation by matching training trajectories"), requiring consolidation.

**Quality:**

4

**Strengths And Weaknesses:**

Strengths
The paper presents EDGE, a generative approach for multimodal dataset distillation that efficiently addresses the computational inefficiencies of existing MTT-based methods. By integrating bidirectional contrastive loss and diversity loss, EDGE enhances image-text correlation and sample diversity, demonstrating superior performance on Flickr30K, COCO, and the large-scale CC3M dataset. Notably, it achieves distillation up to 18 times faster than state-of-the-art methods with 16× lower memory usage, making it feasible for resource-constrained scenarios.
Weaknesses
While EDGE shows significant efficiency gains, the reliance on pre-trained diffusion models like Stable Diffusion may limit its generalizability to specialized domains where model biases exist. The theoretical foundation of the diversity loss lacks explicit justification for matching original dataset distributions, and the study does not validate the method on other multimodal tasks (e.g., video-text).

---

> ### Author Rebuttal · Authors · 2025-07-30
>
> We sincerely thank you for the valuable questions and comments. For the concerns and questions, here are our responses:
> ### **W1. While EDGE shows significant efficiency gains, the reliance on pre-trained diffusion models like Stable Diffusion may limit its generalizability to specialized domains where model biases exist.**
>
> Thank you for your comment. Although our implementation employs a Stable Diffusion backbone that is pre-trained on LAION-5B, we fine-tune the generator jointly with EDGE on every target corpus. Our primary experiments are conducted on Flickr30K and COCO, whose visual styles and caption vocabularies differ from LAION. Before finetuning, Stable Diffusion achieves 3.3 and 1.8 recall@1 on these datasets, respectively; after applying EDGE, the scores rise to 9.9 and 2.8. These gains, together with consistent improvements in MMD, demonstrate that EDGE can adapt a biased or domain-mismatched generator to unfamiliar content libraries with only a modest compute budget.
>
> ### **W2 & Q2 & Q4: The diversity loss lacks explicit justification for matching original dataset distributions. What is the role of the diversity loss in EDGE, and how does it differ from traditional dataset distillation approaches?**
>
> **Table 1.** Feature analysis compared to SOTA methods.
>
> | Methods           | MMD ↓   | NNO ↑   | Entropy ↑ |
> |-------------------|---------|---------|------------|
> | MTT-VL            | 0.0293  | 71.04%  | 7.79       |
> | LoRS              | 0.0261  | 78.11%  | 7.87       |
> | Stable-diffusion  | 0.0286  | 70.55%  | 8.01       |
> | EDGE (ours)       | 0.0242  | 79.68%  | 8.34       |
>
> Thank you for the thoughtful feedback. In EDGE, the diversity loss complements the contrastive objective by explicitly maximizing the pairwise angular distance between distilled image–text embeddings. This design is inspired by the maximum entropy principle: among all low-cardinality datasets that satisfy the contrastive constraints, we prefer the one whose embeddings occupy the largest feasible volume in the representation space. By discouraging co-location of different samples, the loss mitigates the tendency of small distilled sets to collapse onto a few modes and thus helps the synthetic distribution approximate the support of the original data. While the contrastive loss is responsible for aligning corresponding image-text embeddings to ensure semantic consistency, the diversity loss serves as a complementary objective that actively disperses the embeddings of different image-text pairs. In this way, EDGE explicitly encourages inter-sample variation through diversity loss, thereby enhancing generalization and distributional fidelity.
>
> Traditional trajectory-matching-based dataset distillation approaches (MTT-VL, LoRS) usually do not consider the diversity, since they strictly focus on the performance of a specific model architecture, ignoring the evaluation on other models.
>
> Additionally, our feature analysis experiments in Table 1 demonstrate the effectiveness of the diversity loss, where we observe that incorporating it leads to a reduction in MMD and an increase in NNO and Entropy, further supporting its role in promoting broader distributional coverage.
>
> ### **W3: The study does not validate the method on other multimodal tasks.**
>
> **Table 2.** VQA metrics evaluation.
>
> | Methods         | Tifa-score ↑ | DSG score with dependency ↑ | DSG score without dependency ↑ |
> |----------------|------------|----------------------------|------------------------------|
> | LoRS           | 0.856      | 0.852                      | 0.876                        |
> | Stable-diffusion | 0.824    | 0.771                      | 0.805                        |
> | EDGE (ours)    | 0.877      | 0.874                      | 0.894                        |
>
> We appreciate the reviewer’s suggestion. To evaluate EDGE beyond image–text retrieval, we have added experiments on the VQA TIFA [1] and DSG [2] benchmarks in Table 2. These extensions confirm that the advantages conferred by EDGE are not confined to static imagery, and we commit to releasing our code for video distillation to facilitate future research.
>
> [1] TIFA: Accurate and Interpretable Text-to-Image Faithfulness Evaluation with Question Answering. ICCV 2023.
>
> [2] Davidsonian Scene Graph: Improving Reliability in Fine-grained Evaluation for Text-to-Image Generation. ICLR 2024.
>
> ### **Q1: Why is the CC3M dataset particularly challenging for existing dataset distillation methods, and how does EDGE address this?**
>
> Thank you for the question. CC3M poses a substantially tougher testbed than Flickr30K and COCO because of its scale: it contains roughly 3.3 million image–text pairs, compared with 31 K and 123 K images (each with five captions) in Flickr30K and COCO, respectively. Existing distillation approaches rely on trajectory‑matching procedures whose memory and compute demands grow with dataset size. This limitation stems from the objective function of Matching Training Trjectories (MTT) [1], which involves unrolling T steps of SGD using synthetic images and aligning the resulting model weights with a reference point obtained from training on the original dataset. Because this process requires differentiating through T optimization steps, it entails constructing and retaining T gradient computation graphs in memory. As a result, the memory overhead becomes prohibitive when applied to large-scale datasets.
>
> EDGE sidesteps this bottleneck by introducing a lightweight generative distillation strategy that eliminates the need for full‑dataset gradient matching. Instead, it learns a compact generative prior that can be optimized with a small, fixed computational budget, enabling us to distill CC3M efficiently while retaining competitive downstream performance.
>
> [1] Dataset Distillation by Matching Training Trajectories. CVPR 2022.
>
> ### **Q3: What is the purpose of the caption synthesis strategy in EDGE, and how does it work?**
>
> Thank you for the question. The purpose of the caption synthesis strategy in EDGE is to improve performance on the text-to-image retrieval task, which we found to be more challenging than its image-to-text counterpart. As discussed at the beginning of Section 3.4, there exists an inherent asymmetry in retrieval difficulty between these two directions. Specifically, generated images may not always have sufficiently rich or semantically grounded captions, which can hinder the model's ability to learn strong text-to-image alignment.
>
> To address this, we introduce a caption synthesis strategy that augments the captions associated with the generated images. This is achieved by leveraging a pretrained captioning model to generate more informative and diverse textual descriptions, thereby enriching the training data and improving the alignment quality. These synthesized captions supplement the original ones and help the model better associate generated images with a broader set of possible textual queries. Empirically, we observe that this strategy leads to notable improvements in text-to-image retrieval accuracy, as shown in Table 8. We believe this component plays an important role in balancing the retrieval performance across modalities and enhancing the generalization capability of the distilled dataset.
>
> ### **Formatting Concerns**
> Thank you for pointing this out. We will carefully revise all entries to ensure that titles, author lists, publication dates, and venues are complete and accurate.

---

> ### Comment · Area_Chair_xGiR · 2025-08-05
> **Please read the rebuttal**
>
> Dear Reviewer 3o9X,
>
> Please take some time to read the rebuttal and see if you have more questions to the authors. Otherwise, please confirm you have read all reviews and discussions, and update your review accordingly.
>
> Thank you!
>
> AC

---

### Official Review · Reviewer_nNDs · 2025-07-04

**Clarity:** 3
**Significance:** 3
**Originality:** 3
**Rating:** 4
**Confidence:** 3

**Summary:**

This paper introduces EDGE, a method for efficient multimodal dataset distillation using generative models. The authors tackle two main challenges: poor correlation between generated images and text, and lack of diversity in generated samples. Their solution combines a bi-directional contrastive loss with a diversity loss, plus a caption synthesis strategy. Experiments on Flickr30K, COCO, and CC3M show the method works 18× faster than current approaches while maintaining competitive performance.

**Questions:**

See weakness

**Ethical Concerns:**

["NO or VERY MINOR ethics concerns only"]

**Final Justification:**

It addressed my concerns. I will raise the score accordingly

**Limitations:**

See weakness

**Quality:**

3

**Strengths And Weaknesses:**

**Strengths**
1. It successfully distills the large-scale CC3M dataset, which previous methods couldn't handle. The caption synthesis strategy is very practical.

2. The proposed method dramatically reduces computational requirements compared to existing approaches. The ability to distill datasets 18× faster is a substantial contribution.

3. It works consistently across different architectures, unlike baseline methods that show severe performance drops when switching models.

4. The experiments are comprehensive: good ablation studies on different components, caption-per-image ratios, and caption synthesis techniques.

**Weakness**

1. While the paper emphasizes EDGE's efficiency advantages over baseline methods like LoRS (which performs quite well across most setups), it lacks a controlled comparison based on FLOPs. The authors primarily focus on runtime and memory usage comparisons, which can be influenced by implementation details and hardware configurations. A more rigorous efficiency comparison controlling for computational resources would strengthen the claim that EDGE is both efficient and effective.

2. In Table 10, the proposed method performance drops a lot when setting CPI=2 to CPI=5. These results may suggest that the model is bit sensitive and not robust enough to apply to different setups

---

> ### Author Rebuttal · Authors · 2025-07-30
>
> We sincerely thank you for the valuable questions and comments. For the concerns and questions, here are our responses:
> ### **W1: A rigorous efficiency comparison.**
>
> | Method        | FLOPs Required |
> |---------------|----------------|
> | LoRS          | 2528 PFLOPs     |
> | EDGE (ours)   | 149 PFLOPs     |
>
> Thank you for your comment. In the revised manuscript, we have added a new Table, which reports the total floating-point operations required by each method when distilling the COCO dataset used in our study. FLOP counts were computed with the PyTorch profile. All experiments were executed on a single NVIDIA RTX A5000 GPU with CUDA 12.4, and the host system provided 48 CPU cores and 256 GB of RAM.
>
> LoRS consumes 2.5 EFLOPs, whereas EDGE requires only 149 PFLOPs. This metric substantiates the claim that EDGE is intrinsically more compute efficient rather than merely faster due to implementation details.
>
> ### **W2: Explanation of performance drops when setting CPI=2 to CPI=5.**
>
> Thank you for raising this point. In our setup, as CPI increases, the unique images in the generated dataset decrease to preserve the consistency of the total number of image-text pairs. CPI refers to “captions per image.” When we increase CPI from 2 to 5, we intentionally keep the total number of image–text pairs constant for a fair comparison. This means the number of unique images in the generated dataset shrinks from 250 to 100 (2 × 250 = 5 × 100 = 500 pairs). Such a substantial reduction in image diversity naturally lowers performance, so the observed drop reflects the altered dataset composition rather than a lack of robustness in our method.

---

> > ### Comment · Reviewer_nNDs · 2025-08-01
> >
> > It addressed my concerns. I will raise the score accordingly

---

> > > ### Author Response · Authors · 2025-08-02
> > >
> > > Dear reviewer nNDs,
> > >
> > > We would like to express our sincere gratitude to you for acknowledging our work and providing constructive suggestions. We will update the manuscript accordingly. Thanks again for your time and effort in reviewing our work.

---

### Official Review · Reviewer_Hjt5 · 2025-07-05

**Clarity:** 3
**Significance:** 3
**Originality:** 3
**Rating:** 4
**Confidence:** 3

**Summary:**

This paper proposes an efficient multimodal dataset distillation method named EDGE, which is designed to compress large-scale vision-language datasets using generative models, thereby improving model training efficiency and performance. The EDGE method introduces a bidirectional contrastive loss and a diversity loss to enhance the correlation between images and text and to increase sample diversity. Furthermore, a post-training caption synthesis strategy is proposed, which utilizes multimodal large language models (MLLMs) to further boost performance on text-to-image retrieval tasks. Experimental results show that compared to existing methods, EDGE not only achieves superior performance on datasets of varying scales such as Flickr30K, COCO, and CC3M, but also demonstrates significant advantages in speed and computational resource consumption, especially when processing large-scale datasets. This proves its dual advantages in both efficiency and effectiveness.

**Questions:**

1. Could you provide a plot showing the relationship between the performance achieved and the computation time required for the training samples distilled by various methods?

2. Regarding the method's extensibility, could it be applied to other tasks, and what do you see as its most promising application areas?

**Ethical Concerns:**

["NO or VERY MINOR ethics concerns only"]

**Final Justification:**

The authors answer my questions and fix the weakness.

**Limitations:**

Please provide a limitation section.

**Paper Formatting Concerns:**

NA.

**Quality:**

3

**Strengths And Weaknesses:**

### **Strengths:**

1.  This study introduces EDGE, an efficient multimodal dataset distillation method that compresses large-scale vision-language datasets using generative models, thereby improving model training efficiency and performance.

2.  Experimental results demonstrate that, compared to existing methods, EDGE not only achieves superior performance on datasets of varying scales such as Flickr30K, COCO, and CC3M, but also shows significant advantages in speed and computational cost. This proves its dual advantage in both efficiency and effectiveness, especially when processing large-scale datasets.

### **Weaknesses:**

1.  The study lacks a feature analysis of the distilled data samples.

2.  A clearer explanation is needed for the performance gap between the proposed method and LoRS, as seen, for example, in Table 2.

---

> ### Author Rebuttal · Authors · 2025-07-30
>
> We sincerely thank you for the valuable questions and comments. For the concerns and questions, here are our responses:
> ### **W1. The study lacks a feature analysis of the distilled data samples.**
>
> **Table 1.** Feature analysis compared to SOTA methods.
>
> | Methods           | MMD ↓   | NNO ↑   | Entropy ↑ |
> |-------------------|---------|---------|------------|
> | MTT-VL            | 0.0293  | 71.04%  | 7.79       |
> | LoRS              | 0.0261  | 78.11%  | 7.87       |
> | Stable-diffusion  | 0.0286  | 70.55%  | 8.01       |
> | EDGE (ours)       | 0.0242  | 79.68%  | 8.34       |
>
> We appreciate the reviewer’s constructive suggestion and have carried out a comprehensive feature-level evaluation of the distilled dataset. We conducted an extensive analysis in Table 1 that benchmarks our method, against MTT‑VL, LoRS, and a pre-trained Stable Diffusion baseline. Specifically, we include a table reporting Maximum Mean Discrepancy (MMD), Nearest‑Neighbor Overlap (NNO), and Entropy. EDGE achieves the lowest MMD, indicating the closest alignment to the original distribution, while simultaneously attaining the highest NNO  and Entropy, demonstrating superior semantic coverage and lexical richness. These results corroborate the previously reported FID and CLIP‑score improvements, collectively showing that EDGE preserves data fidelity without sacrificing diversity or image–text coherence.
>
> To make the findings more intuitive, we also plotted t‑SNE visualizations that reveal EDGE samples forming compact clusters largely overlapping the original data manifold, whereas competing methods exhibit noticeable drift or mode collapse. We will add these plots and the full numerical results to the revised manuscript.
> ### **W2: A clearer explanation is needed for the performance gap between the proposed method and LoRS.**
> Thank you for your comments. LoRS is fundamentally a trajectory‑matching method [1] that optimizes a synthetic set so that the parameter trajectory taken by a specific network trained on that set mirrors its training trajectory on the original dataset. Such a model-specific method performs outstandingly when using the exact same model architecture, but compared to our method, it has two fatal drawbacks:
>
> 1\. LoRS suffers from cross-model evaluation. Because the LoRS optimisation explicitly relies on gradients from a single architecture, the resulting distilled set is highly specialised: it performs well when evaluated with the same backbone but loses predictive power when the evaluation network changes. As shown in Table 7, when using model architectures that are different from the model used during distillation, the performance of LoRS drops dynamically.
>
> Our proposed EDGE aims to explore the dataset representation regardless of which network is used. From Table 7, it is observed that although LoRS performance drops dynamically, EDGE can still achieve a decent performance when using other network architectures.
>
> 2\. A second factor is the computing budget. Trajectory matching back‑propagates through many unrolled training steps, which becomes prohibitive on large datasets and limits the number of outer‑loop updates LoRS can afford. The trajectory-based distillation methods suffer from a huge computational requirement. As we highlighted in Figure 1 and Table 6, our proposed generative-distillation method EDGE is significantly efficient than LoRS, which is 18 times faster.
>
> In conclusion, the performance gap comes from LoRS sacrificing the performance of cross-model evaluation. Furthermore, our proposed method is more efficient.
>
> [1] Dataset Distillation by Matching Training Trajectories. CVPR 2022.
>
> ### **Q1: Could you provide a plot showing the relationship between the performance achieved and the computation time required for the training samples distilled by various methods?**
>
> **Table 2.** Performance *vs.* GPU hour usage.
>
> | Method       | Validation IR@1 ↑ | Validation TR@1 ↑ | GPU hour |
> |--------------|------------------|------------------|----------|
> | MTT-VL       | 1.8              | 2.5              | 194.1    |
> | LoRS         | 2.5              | 3.6              | 155.6    |
> | EDGE (ours)  | 2.8              | 3.9              | 8.5      |
>
> Thank you for the suggestion. As image uploading is disabled this year, we instead provide Table 2 summarizing the relationship between performance and GPU time for the distilled training samples. The experiments are conducted on NVIDIA RTX A5000 GPUs. We distilled the COCO datasets to 1000 image-text pairs.
>
> From the table, it is clear that our method achieves higher accuracy than state-of-the-art baselines while requiring only a fraction of the computational cost. Specifically, EDGE delivers better performance while using just 5% of the GPU time, demonstrating both superior accuracy and substantial efficiency gains.
>
> ### **Q2: Regarding the method's extensibility, could it be applied to other tasks, and what do you see as its most promising application areas?**
>
> **Table 3.** VQA metrics evaluation.
>
> | Methods           | Tifa-score ↑ | DSG score with dependency ↑ | DSG score without dependency ↑ |
> |-------------------|------------|----------------------------|-------------------------------|
> | LoRS              | 0.856   | 0.852                   | 0.876                   |
> | Stable-diffusion  | 0.824     | 0.771                    | 0.805                      |
> | EDGE (ours)       | 0.877     | 0.874                      | 0.894                         |
>
> We appreciate the reviewer’s interest in our work and address the question of extensibility in two parts: technical adaptability and practical application.
>
> EDGE is designed around a task‑agnostic generative prior and a modality‑aligned objective, so adapting it to new task settings primarily requires substituting the downstream loss or the diffusion backbone. With minor changes, our method can be applied to video-text datasets and other image-text tasks such as VQA. We perform a VQA evaluation on the distilled dataset of COCO with TIFA[1] and DSG[2], compared with other distilled datasets to show the effectiveness of our method in Table 3.
>
> The distilled dataset by our method has multiple promising applications. First of all, the distilled dataset can speed up the training efficiency and protect sensitive content leakage of the original dataset. Besides, the generated dataset can be a part of supplemental data for large generative model training and mLLM model training. The distilled data can also serve as supplemental, diversity‑enhancing supplements for large generative‑model pre‑training or for recent multimodal LLMs, which benefit from additional image–caption pairs yet are often bottlenecked by curation costs.
>
> [1] TIFA: Accurate and Interpretable Text-to-Image Faithfulness Evaluation with Question Answering. ICCV 2023.
>
> [2] Davidsonian Scene Graph: Improving Reliability in Fine-grained Evaluation for Text-to-Image Generation. ICLR 2024.
>
> ### **Limitation of our method**
> Thank you for your thoughtful comment. We acknowledge that our current method may have limitations when applied to highly specialized domains such as medical imaging, where data characteristics can differ significantly from the general datasets used in our evaluation. These domains may involve unique structural patterns or annotation constraints that are not fully captured in our current experiments. We believe future work could explore adapting our approach to such specialized settings by incorporating domain-specific priors or tailoring the optimization process accordingly. We will add a detailed discussion of these limitations in the revised manuscript.

---

> > ### Comment · Reviewer_Hjt5 · 2025-08-02
> >
> > Thanks for your answers, which addressed my concerns. I will raise my score accordingly.

---

> > > ### Author Response · Authors · 2025-08-02
> > >
> > > Dear Reviewer Hjt5,
> > >
> > > Thank you very much for taking the time to review our work. We sincerely appreciate your thoughtful feedback and engagement during the review process.
> > >
> > > Best regards,
> > >
> > > Authors

---

### Author Response · Authors · 2025-08-01
**Follow-up on Discussion**

Dear Reviewers and AC,

We hope you are doing well.

We have submitted our full responses to all reviewer comments. Two reviewers 3o9X and r1pa have expressed positive assessments of our work (both with a score of 5), while Reviewers Hjt5 and nNDs have given a score of 3.

Given this score divergence, feedbacks from Reviewers Hjt5 and nNDs would be impactful in the final decision. As the discussion period is approaching its end on August 6, we would be sincerely grateful if you could kindly share any further concerns or questions you might have.

We are fully available and committed to responding to any additional feedback during this final phase.

Thank you very much for your time and consideration.

Best regards,
The Authors

---

### Decision · Program_Chairs · 2025-09-17

**Decision:**

Accept (poster)

**Comment:**

This paper proposes an efficient and scalable multimodal dataset distillation method that meaningfully advances the field. The key strengths are the combination of generative-based distillation with contrastive and diversity loss functions, as well as a practical caption synthesis step, together with strong experimental validation showing both superior accuracy and dramatically improved efficiency.

The reviewers did note some issues: evaluation could be broadened with more alignment/diversity metrics, qualitative examples would strengthen intuitions about dataset quality, and some claims are phrased more strongly than warranted given the absolute performance numbers. During the rebuttal, the above concerns have been addressed and are promised to be revised accordingly. The reviewers are all positive and vote to accept the paper.